# At Home and on the Brink: U.S. Parents’ Mental Health during COVID-19

**DOI:** 10.3390/ijerph19095586

**Published:** 2022-05-04

**Authors:** Sarah Moreland-Russell, Jason Jabbari, Dan Ferris, Stephen Roll

**Affiliations:** 1Prevention Research Center, Brown School, Washington University in St. Louis, One Brookings Drive, St. Louis, MO 63130, USA; 2Social Policy Institute, Brown School, Washington University in St. Louis, One Brookings Drive, St. Louis, MO 63130, USA; jabbari.jason@wustl.edu (J.J.); dan.ferris@wustl.edu (D.F.); stephen.roll@wustl.edu (S.R.)

**Keywords:** mental health, COVID-19, telework, homeschool, policy

## Abstract

Though the COVID-19 pandemic required significant changes and adaptations for most Americans, parents faced acute challenges as they had to navigate rapidly changing schooling and child care policies requiring their children to spend more time at home. This study examines the effects of COVID-19 school and workplace policies as well as environmental and economic characteristics on parental mental health, worry, hopelessness, and anxiety. Using data from four waves of the Socio-Economic Impacts of COVID-19 Survey and regression analysis, we explore associations between parents’ mental health, worry, hopelessness, and anxiety and school learning environment, child grade and learning disability, employment characteristics, and sociodemographic factors. We find that having a child attend a private school or school with above average instructional quality was associated with better mental health of parents. Hybrid schooling options offering both in-person and online learning was associated with poor parental mental health, as was working from home. Being female or experiencing job or income loss were associated with worse mental health while having older children, a bachelor’s degree, or high income were associated with better mental health. Results can help inform school and workplace family supports as well as opportunities to reduce mental health strains at home from various policy options.

## 1. Introduction

The COVID-19 pandemic required most Americans to change their daily routine, environments, social networks, and general way of life. Policies that focused on social distancing, including telework and distance learning, were enacted across the U.S. to reduce the spread of COVID-19 infections. In 2020, telework became common, as over 70% of American workers whose job responsibilities could mainly be done from home reported a telework arrangement [1]. The pandemic also dramatically shifted the way children were being educated. Over 90% of people in households with school-age children reported their children engaged in some form of distance learning, in which they received their education online [2]. As a result, working parents with school-age children were faced with needing to balance work obligations, protect their family’s health, and play increased roles supporting their child(ren)’s education.

Social distancing policies including telework and distance learning were effective in slowing the spread of infection [3]. However, the pandemic and the mitigation policies adopted in response have resulted in several adverse effects on health. The recent literature [4,5,6] has cited several characteristics associated with poor health among adults during the pandemic: social isolation, strained family relationships, adverse health behaviors, social disorder, and adverse psychosocial states. In addition, economic impacts such as job loss and household hardship [7], disruptions to essential services, and disruptions in education have been reported as negative effects of “distancing” policies [4]. When examining parental mental health in association with school home schooling in Japan, Yamamura et al. (2021) [8] found that school closures led to mothers of younger students suffering from worse mental health. They also observed worse mental health for less-educated mothers who had children attending primary school. Using the Perceived Stress Scale with a sample across 41 countries, Limcaoco et al. (2020) [9] described an increase in parental mental health symptoms due to the COVID-19 pandemic, including anxiety, and reports that women suffered disproportionately from increased anxiety and other psychosocial effects related to increased need to provide childcare.

There is also now a consensus that children learned much less with distance learning in 2020–2021 than during a usual in-person school year and children’s mental health suffered. In a survey completed by the Kaiser Family Foundation [10] in August 2021, almost half (47%) of parents whose kids engaged in distance learning or a mix of in-person and online during the prior school year said they fell behind academically compared with 25% of parents whose children attended schools all or mostly in-person. Viner et al.’s (2022) systematic review of studies from across the globe found school closures and social lockdown during the first COVID-19 wave were associated with adverse mental health symptoms among children and adolescents [11]. Several studies [12,13] report that the impact of the pandemic on K–12 student learning was also significant across all ages, leaving students behind in several subjects, including mathematics and reading. Douglas et al. (2020) [4] reported that these effects are even more daunting for families with low incomes or limited employment options, families who face greater health risks, and families who face inequities in access to educational and health resources, deepening the racial and ethnic divide in educational attainment and overall health and wellbeing.

Lombardi et al. (2010) offer a conceptual model that describes the interaction of childhood learning environment, family wellbeing, overall health including mental health, and family support [14]. This model posits that policy changes in one area (i.e., learning environment or family wellbeing) should not be viewed in isolation, but as affecting all components. Therefore, policies that affect learning environments (online learning) can impact health and family wellbeing. In turn, policies that affect family wellbeing (loss of job, working from home, etc.) can also adversely affect overall child health. As noted above, populations within certain demographics or social circumstances may experience more profound adverse effects.

While this model was originally developed in regard to early care education, it offers a useful application for consideration and contextualizing of COVID-19 and policy disruptions across learning environments from early care to high school.

The purpose of this study is to examine the effects of school and workplace policies enacted during the COVID-19 pandemic on parental mental health, building from the Lombardi model [14] of policy interactions affecting children and families as a conceptual basis. Specifically, we present how school policy (in-person, online only, mix of in-person and online, choice of in-person and online), workplace policy (ability to work from home and inability to work from home), and other economic (income) and environmental characteristics associate with parental mental health. By accounting for pre-pandemic measures of mental health, we are able to approximate the change in mental health as a result of changes in policy between July 2020 and June 2021—leveraging nationally representative longitudinal data from the Socio-Economic Impacts of COVID-19 Survey across four waves of collection.

## 2. Materials and Methods

### 2.1. Data and Sample

The data analyzed in this study come from four waves of the Socio-Economic Impacts of COVID-19 Survey [15]. The survey captures a diverse array of household experiences during the pandemic through a detailed set of questions concerning both adults and children in the household [15]. This survey was administered through a large online panel provider and used quota sampling to obtain a sample that approximates the characteristics of the U.S. population in terms of age, gender, race/ethnicity, and income [15]. In doing so, panel members were invited into the survey via email, and upon completion of the survey, were provided incentives [15]. The survey uses a hybrid design, which combines longitudinal cross-sectional data sampling strategies, recruiting new respondents in each wave, while also allowing for re-contacts of prior-wave respondents [15]. The analyses presented in this paper used Waves 2 through 5 of the survey, which were administered at quarterly intervals between July 2020 and June 2021. Total response rates for these surveys ranged between 7% and 14% [16]. Across each wave, between 4893 and 5051 respondents completed the surveys. Re-contact rates ranged between 33% and 63% across the four survey waves. Variables that were believed not to vary in subsequent waves, such as having a child with a diagnosed learning disability, were not re-collected for survey repeaters; all other variables were re-collected in subsequent waves. Although the university’s institutional review board established that this study was not human subjects research, researchers still obtained informed consent from participants prior to administering the survey.

Rather than asking questions about every child in the household, the Socio-Economic Impacts of COVID-19 Survey asked parents about one of their children—the “reference child”. The survey identified this child by first asking parents how many children they had in their household as well as the age of each child. Then the survey randomly selected one of these children and asked their parent (i.e., the respondent) questions about that specific child. In practice, this means that parents with one child were always asked questions about that child, parents of two children will be asked about their oldest child 50% of the time, and so on.

The focus of this study concerns working parents of school-aged children. As such, we restricted the sample to parents who reported that they were employed during a given survey wave, and who reported that their reference child was enrolled in school (*n* = 2318). In addition, we used listwise deletion to exclude respondents who did not provide a response to the items used in the analysis, resulting in an analytic sample of 1913 participants. We also included additional responses from the same respondent over subsequent survey waves. In total, 64% of respondents only completed one wave of the survey, 21% completed two waves, 11% completed three waves, and 4% completed four waves. When these responses are included, the final number of observations in our analytic sample increased to 3004. Thus, through a cross-sectional approach with four waves of data, we were able to observe some of the 1913 participants multiple times, resulting in 3004 observations.

### 2.2. Measures

The primary dependent variables in this study include four measures of mental health. The first measure is based on a general self-reported question asking respondents “How is your mental health currently?” (1 = excellent; 2 = very good; 3 = good; 4 = fair; 5 = poor). We also included a pre-pandemic measure of mental health, asking respondents to describe their mental health prior to the pandemic. The next three dependent variables were derived from the Patient Health Questionnaire for Depression and Anxiety (PHQ-4) scale [17] and captured:Frequency of worry: “How often have you not been able to stop worrying in the past 3 months?”Frequency of hopelessness: “How often have you felt down or hopeless in the past 3 months?”Frequency of anxiety: “How often have you felt nervous or anxious in the past 3 months?”

Each of these variables include four response categories (1 = not at all; 2 = several days; 3 = more than half the days; 4 = nearly every day).

The independent variables in this study include an array of measures capturing educational, child, employment, sociodemographic, and household characteristics. Educational variables include school type (public, charter, private, home-schooled), school plan (in-person, online only, mix of in-person and online, choice of in-person and online), and parents’ assessment of the school’s instructional quality (below average, average, above average). Child characteristics include child grade (kindergarten, 1st–2nd grade, 3rd–5th grade, 6th–8th grade, 9th–12th grade) and the presence of a learning disability in the child. Employment characteristics include respondent’s employment type (full-time, part-time, not working), spousal employment (full-time, part-time, not working, single (here, having a spouse or partner was combined with spousal employment)), and ability to work from home (never, occasionally, always). Sociodemographic characteristics include respondent’s age, gender, race/ethnicity, and income (0–50% of Area Median Income (AMI); 50–80% AMI; 80–120% AMI; 120%+ AMI). Household characteristics include the number of children in the household, primary language spoken at home, resources at home (broadband internet, online learning tools, a quiet place to study, an adult to help with learning tasks), and urbanicity (metro, urban, rural). Additionally, we controlled for respondents’ self-reported mental health prior to COVID-19 as well as the survey wave. Finally, we also incorporated county-level COVID-19 cases and U.S. state of residence in our propensity score estimation models.

### 2.3. Methods

To estimate the relationships between parents’ mental health and our independent variables of interest, we used an ordered logistic model that accounts for the ordinal scales used in each of our outcome variables. Additionally, because our data are longitudinal, with some individuals being observed in multiple waves, we clustered standard errors at the individual level. Finally, as one of the main factors affecting the daily lives of parents during the pandemic—school learning plans—was prone to selection bias, we employed a propensity score weighting strategy. Propensity score weights use the inverse probability of receiving a treatment to balance groups on observed characteristics [18]. While propensity score weighting typically involves a binary treatment, recent extensions from Imbens (2000) [19] and McCaffrey and colleagues (2013) [20] have extended this strategy for multinomial treatments (i.e., treatments with more than two conditions).

Moreover, since model misspecification errors have been shown to bias estimates of treatment effects, especially when using propensity scores [21], we utilized generalized boosted modeling (GBM) to estimate propensity scores. Nonparametric modeling approaches such as GBM have been shown to reduce the chance of these errors [22]. Specifically, GBM utilizes automated, data adaptive modeling algorithms and machine learning techniques to generate propensity scores that balance all modeled covariates across each level of our treatment variable. In estimating the propensity score weights, we used the TWANG—Toolkit for Weighting and Analysis of Non-equivalent Groups—package [23] in Stata.

When considering the factors that can affect school learning plans, we balanced our four treatment conditions—(a) in-person only learning; (b) online-only learning; (c) a mix of in-person and online learning; and (d) a choice between in-person and online learning—on school context (traditional public, charter, and private), geographic context (urbanicity), health context (county-level COVID-19 cases), and policy context (state of residence). After effectively balancing households on school plans, we applied these multinomial propensity score (MNPS) weights in our ordered logistic outcome models.

## 3. Results

In Table 1, we describe our sample. The mean mental health score was 2.336, which is slightly worse than a rating of “very good”, as well as slightly worse than individual’s pre-pandemic rating of mental health (mean = 2.196). Frequencies of worry (mean = 1.916), hopelessness (mean = 1.882), and anxiety (mean = 2.001) indicated that these feelings occurred roughly several days during the past three months for respondents. The majority of the respondents had children who attended traditional public schools (62.7%) and had slightly above the average rating of school quality during the pandemic (mean = 2.390). A plurality of respondents had children whose main learning modality was completely online (37.1%), as well as children who were in high school (grades 9–12) (32.8%). A plurality of households had one child (47%). The average age of respondents was 39, and the majority of respondents were male (54.9%), were White (59%), lived in households that primarily spoke English (95.6%), and lived in metropolitan areas (90.6%). The majority of households had a bachelor’s degree or higher, and a plurality of respondents were low-income (0–50% of AMI). Overall, 36.4% of households lost their job or income due to COVID-19, and the majority of respondents were working full-time (85.9%) and had spouses that were working full-time (58.9%) and could not work from home (56.4%). While over 60% of respondents lived in households with broadband access (65.7%), online learning tools (65.6%), and quiet places to study (60.9%), less than half or respondents (44.5%) lived in households with an adult who could help their children with learning.

In Model 1 (Table 2), we explored the associations between educational, child, employment, sociodemographic, and household characteristics and parental mental health. Positive coefficients are interpreted as being associated with worse mental health, while negative coefficients are interpreted as being associated with better mental health. Starting with factors that are associated with better mental health, we see that having a child attend a private school (*b* = −0.382 **), having an above average perception of school quality (*b* = −0.928 ***), having a middle school (grades 6th–8th) child (*b* = −0.466 *), having a child with a diagnosed learning disability (*b* = −0.387 **), having a bachelor’s degree (*b* = −0.452 **), having high household income (*b* = −0.322 *), and having online learning tools (*b* = −0.359 ***) at home were all associated with better mental health. Conversely, being offered either a mix of in-person and online schooling (a hybrid model) (*b* = 0.252 *) or a choice between online and in-person schooling (an either/or model) (*b* = 0.618 ***), identifying as female (*b* = 0.340 **), identifying as Hispanic (*b* = 0.391 **), experiencing a job or income loss due to COVID-19 (*b* = 0.261 *), working part-time (*b* = 0.392 *), or not working at all (*b* = 1.229 **)—when compared to working full-time, working from home (*b* = 0.639 ***), and being single (*b* = 0.437 *) were all associated with poor mental health.

In Model 2 (Table 2), we added a pre-COVID-19 measure of mental health, which was significantly related to current mental health (*b* = −0.382 **). While most significant relationships remained, there were a few notable changes. First, a mix of in-person and online learning, having a high income, and having an online learning tool at home were no longer significantly associated with mental health. Second, having a high school age child (grades 9–12) was significantly associated with better mental health (*b* = −0.409 *), while having moderate (as opposed to low) income (*b* = 0.367 **) and having a spouse who worked part-time (*b* = 0.387 ****) or did not work at all (compared to working full-time) (*b* = 0.284 *) was significantly associated with worse mental health.

In Table 3, we ran additional models for increased worry (Model 3), hopelessness (Model 4), and anxiety (Model 5). In all of these models, we controlled for prior (i.e., pre-COVID-19) measures of mental health, which was significantly associated with all three outcomes (worry: *b* = 0.442 ***; hopelessness: *b* = 0.384 ***; anxiety: *b* = 0.359 ***). Starting with increased worry (Model 3), we see that having your child attend a charter school (*b* = 0.473 **) or be home-schooled (*b* = 0.507 **), having a child with a diagnosed learning disability (*b* = 0.658 ***), and experiencing job or income loss during the pandemic (*b* = 0.696 ***) were all associated with increased worry. Conversely, being older (*b* = −0.036 ***), being able to work from home (*b* = −0.359 **) occasionally and all the time (*b* = −0.569 ***), and having an adult at home to assist your child’s learning (*b* = −0.579 ***) were all associated with decreased worry.

Moving on to increased hopelessness (Model 4), we see many of the same associations. Specifically, having a child home-schooled (*b* = 0.442 **), having a child with a diagnosed learning disability (*b* = 0.690 ***), and experiencing job or income loss during the pandemic (*b* = 0.898 ***) were all associated with increased hopelessness, while being older (*b* = −0.037 ***), being able to work from home occasionally (*b* = −0.283 *) or all the time (*b* = −0.547 ***), and having an adult at home to assist your child’s learning were all associated with decreased hopelessness (*b* = −0.579 ***). Furthermore, we see that having average (as opposed to poor) school quality (*b* = −0.514 ***), having three or more children (*b* = −0.386 *), identifying as female (*b* = −0.324 **), identifying as Black (*b* = −0.330 *), and having broadband internet at home (*b* = −0.202 *) were significantly associated with decreased hopelessness. However, attending a charter school was not significantly associated with increased hopelessness.

Finally, many of the same associations with increased hopelessness were also associated with increased anxiety (Model 5). Having a child home-schooled (*b* = 0.404 *), having a child with a diagnosed learning disability (*b* = 0.733 ***), and experiencing job or income loss during the pandemic (*b* = 0.783 ***) were all associated with increased anxiety, while having average (as opposed to poor) school quality (*b* = −0.418 **), being older (*b* = −0.032 ***), identifying as Black (*b* = −0.472 **), being able to work from home occasionally (*b* = −0.302 **) and all the time (*b* = −0.461 ***), and having an adult at home to assist your child’s learning (*b* = −0.500 ***) were all associated with decreased anxiety. Furthermore, we see that identifying as an “Other” racial/ethnic group (*b* = −1.053 **) was significantly associated with decreased anxiety, while having a 1st or 2nd grade child (*b* = 0.382 *) and having a middle school (grades 6th–8th) child (*b* = 0.468 **) were significantly associated with increased anxiety. However, having three or more children, identifying as female, and having broadband internet at home were not significantly associated with increased anxiety.

## 4. Discussion

This study is the first to present the interaction of school and workplace policies and other environmental and economic effects on the mental health, hopelessness, anxiety, and worry of working parents in the U.S. It is important to consider how the findings from this study can inform the design of future policies to help support parents’ mental health during times of crisis and other events that may disrupt their lives or those of their children. In regard to school policy and student characteristics, our findings suggest that having a child attend a private school, having a child of middle school (grades 6th–8th) age, having a child with a learning disability, and having a child attend a school with above average instructional quality are associated with better parental mental health. In contrast, having a child with a mix of in-person and online schooling or a choice between online and in-person schooling are associated with worse parental mental health. These findings suggest that parents with the means to afford private school or who are able to send their child to a school with higher instructional quality fared better in regard to mental health during the pandemic. These findings also highlight the importance of keeping students in school to ensure consistent and constant learning and to maintain parental mental health. In addition, this may also indicate that providing parents a choice in how their children attend school (e.g., in-person or remote learning) is not necessarily good for families, and it may be better for schools to focus on providing high-quality education and student supports within one learning modality, rather than trying to handle multiple modalities simultaneously.

For parent characteristics, we see that respondent’s gender—specifically identifying as female—was associated with poor mental health. This finding reinforces the disproportionate caretaking responsibilities for women in the U.S. and the associated greater burden in overcoming challenges related to the pandemic, especially those who have younger children in public school, a child with a learning disability, and who are distance learning. Other studies, though international in scope, have reported on parental mental health during the COVID-19 pandemic. Our findings, along with findings from other studies, suggest the need for workplace policies and social norms that support increased share of childcare across both parents and genders. Notably, proposed federal legislation and policies such as the ‘Moms Matter Act’ would provide targeted funds and supports for maternal mental health. Research has shown that previous policy change such as expansion of the Earned Income Tax Credit had notable positive effects on parent mental health, particularly for mothers. Other policy efforts to improve access to services (e.g., strengthening mental health parity and insurance coverage) or to address strains felt by parents reflected in our results through investment in high-quality, universal pre-K, child care subsidies, and paid family leave have also emerged at both state and federal levels [24].

Although there were some inconsistences across predictors of mental health and worry, hopelessness, and anxiety, such as working from home and having a child with a learning disability, it is important to note that these measures do operate on different time continuums. While mental health was captured ‘in the moment’, worry, hopelessness, and anxiety stretch across the previous three months. Furthermore, while the mental health measure is broad and subjective in nature, worry, hopelessness, and anxiety measures ask about specific experiences that are conceptually different and more concrete. For example, working from home may be mentally draining for working parents, but at the same time, parents may be less worried about contracting COVID-19. Similarly, having a child with a learning disability may cause parents to worry more about their child contracting COVID-19 as these parents may be more likely to seek additional services outside the home. At the same time, the pandemic may have provided these parents with increased family bonding time, which could have improved their mental health. In each case, future research should further explore these differences across mental health and frequency of worry, hopelessness, and anxiety.

While the COVID-19 pandemic abruptly upended normal work routines, it also caused an examination of the potential for workers to engage in online or virtual environments not only during a pandemic, but during “normal” times. Our findings suggest that having the flexibility to work from home causes less worry, but only if the respondent is not also required to care for children during work time. Other studies [25,26] have reported the emergence of work-from-home policies and note the variation of policies across and within industries with respect to how COVID-19 has required a constant reinvention of operations and affected the demands and resources of different jobs, affecting overall wellbeing of employees.

Finally, in regard to work policy and economic-related characteristics, working from home or part-time and job or income loss due to COVID-19 was associated with poor parental mental health; however, it is important to note that these findings are for parents only—who were often forced to work and look after their children while at home during the pandemic; future studies should explore the impact of flexible work schedules on non-parents.

### Limitations

There are three main limitations to this study. First, while we were able to capture retrospective measures, such as respondent’s pre-pandemic perceptions of mental health, we were not able to examine within-person changes over time through a longitudinal design, which limits our ability to make causal inferences. Second, although we were able to account for observable characteristics associated with selection into school learning plans through propensity score weights, it is possible that unobserved characteristics were still influencing our results. Finally, as this survey was based on perceptions of the respondent, our results are prone to bias. Future research should seek to merge in administrative data, such as data from children’s schools, to better understand the relationship between the characteristics of educational institutions and parents’ mental health.

## 5. Conclusions

This study sought to understand the immediate, real-time effects that the COVID-19 pandemic had on foundations of daily life for many. Families, schools, and employers have navigated remarkable challenges in response to the COVID-19 pandemic. The experience of parents in particular, and the resulting mental health strains on these parents, warrants significant attention from researchers, policymakers, and educators and employers alike. The implications of this research point to a number of policy considerations directed toward both pandemic recovery and future planning efforts. Recovery from the pandemic will involve concerted efforts to improve mental health, overall wellbeing, and job and income stability and in “catching up” students socially and academically.

While mental health has taken center stage in U.S. public health, continued research is needed to understand the true toll of the pandemic on families and children. Our findings suggest a need for strategies that focus on providing resources and enhancing community support, especially for women who suffered disproportionately during the pandemic. Parental mental health could also benefit from future policies that help those students who have fallen behind, with increased efforts in areas that serve those students most in need. States and districts have a critical role to play in marshaling Elementary and Secondary School Emergency Relief (ESSER) funding into evidence-based and sustainable initiatives that keep students in school safely while mitigating COVID-19 learning loss and parental distress.

Future planning efforts should also consider a number of findings from this research. While a significant amount of workplace and school policy decision making was advanced by public health guidance, in many cases those decisions were made without acknowledging the large strain being put on working parents, who had to act as teachers and child care workers, while navigating a rapidly changing pandemic and still trying to make ends meet. In the future, school policies and procedures should be developed considering these strains and offer consistent and high-quality instruction for all students. Future workplace policy planning should also actively support the health and wellbeing of employees by listening to their needs and recognizing the significant and unique challenges that workers with caretaking responsibilities face.

For some families, daily life may begin to resemble what it was prior to the pandemic. Many, however, will still be adapting and responding to the idea of living, working, and learning in a “new normal”. The pandemic impacts on parents’ mental health in either scenario will persist, and policymakers will continue to be challenged with how best to respond. We hope that what our research reveals about the intersections between parents’ mental health and their employment, education, and socio-economic situation, can help to inform decisions, flexibilities, and supports across a wide range of school and workplace policies.

## Figures and Tables

**Table 1 ijerph-19-05586-t001:** Study sample.

	Mean	sd	Min	Max
Current Mental Health	2.336	1.107	1.000	5.000
Mental Health (pre-COVID-19)	2.196	0.999	1.000	5.000
Frequency of Worry	1.916	0.998	1.000	4.000
Frequency of Hopelessness	1.882	1.001	1.000	4.000
Frequency of Anxiety	2.001	0.998	1.000	4.000
School Type: Public	0.627		0.000	1.000
School Type: Charter	0.090		0.000	1.000
School Type: Private	0.206		0.000	1.000
School Type: Home-Schooled	0.077		0.000	1.000
School Plan: In-Person Only	0.186		0.000	1.000
School Plan: Online Only	0.371		0.000	1.000
School Plan: Mix of In-Person and Online	0.354		0.000	1.000
School Plan: Choice of In-Person and Online	0.089		0.000	1.000
School’s instructional quality during the pandemic	2.390	0.708	1.000	3.000
Child Grade: Kindergarten	0.088		0.000	1.000
Child Grade: 1st–2nd Grade	0.182		0.000	1.000
Child Grade: 3rd–5th Grade	0.208		0.000	1.000
Child Grade: 6th–8th Grade	0.194		0.000	1.000
Child Grade: 9th–12th Grade	0.328		0.000	1.000
Child has a diagnosed learning disability	0.194		0.000	1.000
Household Children: 1	0.470		0.000	1.000
Household Children: 2	0.406		0.000	1.000
Household Children: 3+	0.124		0.000	1.000
Respondent’s Age	39.111	9.749	18.000	72.000
Respondent’s Gender (Female)	0.451		0.000	1.000
Race/Ethnicity: White	0.590		0.000	1.000
Race/Ethnicity: Black	0.114		0.000	1.000
Race/Ethnicity: Asian	0.061		0.000	1.000
Race/Ethnicity: Hispanic	0.215		0.000	1.000
Race/Ethnicity: Other	0.020		0.000	1.000
English is the primary language spoken in respondent’s home	0.956		0.000	1.000
Education Level: Bachelor’s degree or higher	0.796		0.000	1.000
Income: Low	0.408		0.000	1.000
Income: Moderate	0.219		0.000	1.000
Income: Middle	0.185		0.000	1.000
Income: High	0.188		0.000	1.000
Income/job loss due to COVID-19	0.364		0.000	1.000
Employment: Working Full-Time	0.859		0.000	1.000
Employment: Working Part-Time	0.127		0.000	1.000
Employment: Not Working	0.014		0.000	1.000
Spouse Employment: Working Full-Time	0.589		0.000	1.000
Spouse Employment: Working Part-Time	0.102		0.000	1.000
Spouse Employment: Not Working	0.152		0.000	1.000
Spouse Employment: Single	0.157		0.000	1.000
Work from Home: No	0.564		0.000	1.000
Work from Home: Occasionally	0.219		0.000	1.000
Work from Home: Yes	0.218		0.000	1.000
Resources from Home: Broadband internet connection	0.657		0.000	1.000
Resources from Home: Online learning tool	0.656		0.000	1.000
Resources from Home: Quiet space to study	0.609		0.000	1.000
Resources from Home: An adult who helps with learning	0.445		0.000	1.000
Urbanicity: Metro Area	0.906		0.000	1.000
Urbanicity: Urban Area	0.090		0.000	1.000
Urbanicity: Rural Area	0.004		0.000	1.000
Survey Wave: 2	0.238		0.000	1.000
Survey Wave: 3	0.262		0.000	1.000
Survey Wave: 4	0.239		0.000	1.000
Survey Wave: 5	0.261		0.000	1.000
Observations	3004			

**Table 2 ijerph-19-05586-t002:** Parent mental health during COVID-19: controlling for prior mental health (ordered logistic multivariate regression models).

	(1)		(2)	
	Poor Mental Health		Poor Mental Health	
School Type: Charter	−0.157	(0.198)	−0.186	(0.203)
School Type: Private	−0.382 **	(0.124)	−0.350 **	(0.115)
School Type: Home-Schooled	−0.067	(0.202)	−0.314	(0.194)
School Plan: Online Only	0.172	(0.123)	−0.005	(0.120)
School Plan: Mix of In-Person and Online	0.252 *	(0.119)	0.057	(0.112)
School Plan: Choice of In-Person and Online	0.618 ***	(0.172)	0.379 *	(0.158)
School Quality: Average	0.020	(0.163)	−0.053	(0.150)
School Quality: Good	−0.928 ***	(0.162)	−0.623 ***	(0.148)
Child Grade: 1st–2nd Grade	−0.155	(0.193)	−0.219	(0.190)
Child Grade: 3rd–5th Grade	−0.303	(0.183)	−0.302	(0.181)
Child Grade: 6th–8th Grade	−0.466 *	(0.184)	−0.579 **	(0.187)
Child Grade: 9th–12th Grade	−0.331	(0.188)	−0.409 *	(0.180)
Learning Disability: Yes	−0.387 **	(0.133)	−0.394 **	(0.139)
Household Children: 2	−0.126	(0.107)	−0.128	(0.104)
Household Children: 3+	0.015	(0.163)	−0.043	(0.157)
Respondent’s Age	0.000	(0.007)	−0.003	(0.006)
Gender: Female	0.340 **	(0.121)	0.230 *	(0.108)
Race/Ethnicity: Black	0.204	(0.147)	0.258	(0.156)
Race/Ethnicity: Asian	0.248	(0.186)	0.173	(0.146)
Race/Ethnicity: Hispanic	0.391 **	(0.151)	0.367 **	(0.133)
Race/Ethnicity: Other	0.132	(0.355)	−0.242	(0.397)
Primary Language: English	0.552	(0.283)	0.332	(0.226)
Bachelor’s Degree: Yes	−0.452 **	(0.141)	−0.373 **	(0.135)
Income: Moderate	0.033	(0.132)	0.367 **	(0.124)
Income: Middle	−0.073	(0.153)	0.230	(0.139)
Income: High	−0.322 *	(0.156)	−0.022	(0.152)
Job/Income Loss: Yes	0.261 *	(0.109)	0.215 *	(0.108)
Employment: Working Part-Time	0.392 *	(0.164)	0.347 *	(0.174)
Employment: Not Working	1.229 **	(0.390)	1.428 **	(0.442)
Spouse Employment: Working Part-Time	0.223	(0.154)	0.387 **	(0.150)
Spouse Employment: Not Working	0.054	(0.147)	0.284 *	(0.136)
Spouse Employment: Single	0.437 *	(0.171)	0.348 *	(0.156)
Work from Home: Occasionally	0.085	(0.112)	0.009	(0.115)
Work from Home: Yes	0.639 ***	(0.135)	0.301 *	(0.118)
Broadband at Home: Yes	−0.076	(0.108)	−0.022	(0.112)
Online Learning Tool at Home: Yes	−0.359 ***	(0.106)	−0.093	(0.099)
Quiet Place to Study at Home at Home: Yes	0.005	(0.105)	0.196	(0.103)
Adult Learning Assistance at Home: Yes	0.083	(0.110)	0.028	(0.105)
Urbanicity: Urban Area	0.162	(0.180)	−0.043	(0.142)
Urbanicity: Rural Area	−0.075	(0.900)	−0.621	(0.516)
Mental Health Before Pandemic			1.691 ***	(0.081)
cut1	−1.187 *	(0.476)	1.951 ***	(0.444)
cut2	0.294	(0.475)	4.051 ***	(0.467)
cut3	1.917 ***	(0.477)	6.355 ***	(0.504)
cut4	3.541 ***	(0.502)	8.457 ***	(0.561)
Observations	3004		3004	
Pseudo *R*^2^	0.085		0.251	

*Notes*: Standard errors in parentheses; wave variable not shown; standard errors are clustered at individual level; propensity score weights applied. *Reference categories*: …School Type (Public); School Plan (In-Person Only); School Quality (Poor) …Child Grade (Kindergarten); Learning Disability (No) …Household Children (0); Gender (Male); Race (NH White); Primary Language (Not English); …Bachelor’s Degree (No); Income (Low); Job/Income Loss (No); …Employment (Working Full-Time); Spouse Employment (Working Full-Time); …Work from Home (No); …Broadband at Home, Online Learning Tool at Home, Quiet Place to Study at Home (No); …Adult Learning Assistance at Home (No); …Urbanicity (Metro Area); …Survey Wave (2); * *p* < 0.05, ** *p* < 0.01, *** *p* < 0.001.

**Table 3 ijerph-19-05586-t003:** Worry, hopelessness, and anxiety during COVID-19 (ordered logistic multivariate regression models).

	(1)		(2)		(3)	
	Frequency of Worry		Frequency of Hopelessness		Frequency of Anxiety	
Mental Health Before Pandemic	0.442 ***	(0.053)	0.384 ***	(0.054)	0.359 ***	(0.055)
School Type: Charter	0.473 **	(0.162)	0.326	(0.174)	0.287	(0.167)
School Type: Private	0.146	(0.122)	0.124	(0.126)	0.153	(0.123)
School Type: Home-Schooled	0.507 **	(0.190)	0.442 **	(0.158)	0.404 *	(0.161)
School Plan: Online Only	0.105	(0.128)	0.237	(0.129)	0.073	(0.123)
School Plan: In-Person and Online	0.000	(0.126)	−0.042	(0.124)	−0.065	(0.117)
School Plan: In-Person or Online	−0.005	(0.170)	0.080	(0.171)	−0.183	(0.165)
School Quality: Average	−0.273	(0.139)	−0.514 ***	(0.147)	−0.418 **	(0.139)
School Quality: Good	−0.112	(0.131)	−0.210	(0.145)	−0.201	(0.139)
Child Grade: 1st–2nd Grade	0.091	(0.182)	0.168	(0.176)	0.382 *	(0.174)
Child Grade: 3rd–5th Grade	0.319	(0.182)	0.256	(0.173)	0.325	(0.167)
Child Grade: 6th–8th Grade	0.306	(0.183)	0.286	(0.180)	0.468 **	(0.180)
Child Grade: 9th–12th Grade	0.136	(0.183)	0.049	(0.181)	0.156	(0.182)
Learning Disability: Yes	0.658 ***	(0.116)	0.690 ***	(0.124)	0.733 ***	(0.124)
Household Children: 2	−0.188	(0.102)	−0.186	(0.107)	−0.117	(0.102)
Household Children: 3+	−0.292	(0.163)	−0.386 *	(0.156)	−0.234	(0.163)
Respondent’s Age	−0.036 ***	(0.006)	−0.037 ***	(0.006)	−0.032 ***	(0.006)
Gender: Female	−0.065	(0.110)	−0.324 **	(0.113)	0.027	(0.110)
Race/Ethnicity: Black	−0.242	(0.173)	−0.330 *	(0.149)	−0.472 **	(0.155)
Race/Ethnicity: Asian	−0.380	(0.218)	−0.088	(0.210)	−0.349	(0.219)
Race/Ethnicity: Hispanic	0.154	(0.134)	0.267	(0.139)	0.037	(0.136)
Race/Ethnicity: Other	−0.787	(0.427)	−0.174	(0.416)	−1.053 **	(0.390)
Primary Language: English	−0.001	(0.234)	0.136	(0.240)	0.097	(0.236)
Bachelor’s Degree: Yes	0.186	(0.141)	−0.022	(0.142)	0.202	(0.140)
Income: Moderate	−0.165	(0.136)	−0.220	(0.137)	−0.067	(0.132)
Income: Middle	−0.179	(0.149)	−0.164	(0.161)	−0.090	(0.150)
Income: High	−0.209	(0.144)	−0.074	(0.152)	−0.162	(0.143)
Job/Income Loss: Yes	0.696 ***	(0.104)	0.898 ***	(0.107)	0.783 ***	(0.105)
Employment: Working Part-Time	0.106	(0.141)	0.209	(0.146)	0.165	(0.137)
Employment: Not Working	−0.224	(0.446)	−0.114	(0.526)	−0.047	(0.379)
Spouse Employment: Part-Time	−0.045	(0.157)	−0.112	(0.164)	0.035	(0.153)
Spouse Employment: Not Working	−0.310	(0.163)	−0.236	(0.156)	−0.224	(0.154)
Spouse Employment: Single	−0.134	(0.159)	−0.134	(0.160)	−0.242	(0.160)
Work from Home: Occasionally	−0.359 **	(0.115)	−0.283 *	(0.114)	−0.302 **	(0.107)
Work from Home: Yes	−0.569 ***	(0.133)	−0.547 ***	(0.143)	−0.461 ***	(0.128)
Broadband at Home: Yes	−0.028	(0.104)	−0.202 *	(0.103)	−0.043	(0.105)
Online Learning Tool at Home	0.057	(0.109)	−0.061	(0.106)	0.186	(0.105)
Quiet Place to Study at Home	−0.134	(0.102)	−0.111	(0.103)	0.008	(0.102)
Adult Learning Assistance at Home	−0.579 ***	(0.112)	−0.579 ***	(0.110)	−0.500 ***	(0.108)
Urbanicity: Urban Area	0.236	(0.155)	0.284	(0.176)	0.260	(0.166)
Urbanicity: Rural Area	0.760	(0.721)	0.078	(1.263)	−0.421	(0.588)
cut1	−0.989 *	(0.425)	−1.323 **	(0.435)	−0.974 *	(0.430)
cut2	0.504	(0.427)	0.164	(0.436)	0.801	(0.428)
cut3	2.122 ***	(0.440)	1.785 ***	(0.436)	2.112 ***	(0.435)
Observations	2997		2987		3004	
Pseudo *R*^2^	0.124		0.142		0.112	

*Notes*: Standard errors in parentheses; wave variable not shown; standard errors are clustered at individual level; propensity score weights applied. *Reference categories*: …School Type (Public); School Plan (In-Person Only); School Quality (Poor); …Child Grade (Kindergarten); Learning Disability (No); …Household Children (0); Gender (Male); Race (NH White); Primary Language (Not English); …Bachelor’s Degree (No); Income (Low); Job/Income Loss (No); …Employment (Working Full-Time); Spouse Employment (Working Full-Time); …Work from Home (No); …Broadband at Home, Online Learning Tool at Home, Quiet Place to Study at Home (No); …Adult Learning Assistance at Home (No); …Urbanicity (Metro Area); …Survey Wave (2); * *p* < 0.05, ** *p* < 0.01, *** *p* < 0.001.

## Data Availability

The authors may provide data upon request.

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
