# Peer review of "At Home and on the Brink: U.S. Parents’ Mental Health during COVID-19"

_ijerph, 2022, doi:10.3390/ijerph19095586_

Round 1

Reviewer 1 Report

This paper analyzes the mental health of parents in the Corona situation. It is necessary to consider the following points.
First, this study give focuse on the topic of mental health of parents with educational burden in the COVID-19 situation, but the theoretical part is weak. I hope that theoretical discussions related to mental health, quality of life, and education burden on parents, and mental health in the context of COVID-19 will be added. In particular, as many empirical studies have been conducted on worry and anxiety, which play the role of dependent variables, a systematic theoretical review should be made on them.
Second, the analysis is being conducted focusing on simple frequency analysis. It is necessary to conduct a regression analysis in which mental health is set as a dependent variable and education type, employment status and other variables are set as independent variables.
Third, information on the sampling method or basic characteristics of the sample should be provided.

Reviewer 2 Report

Thanks for allowing me to review this manuscript.

Below are my comments for the authors to consider for further improving the quality of the manuscript:

A. It is suggested to update the title to inform readers that the study focused on parents in the United States. It is also suggested to revise the abstract to include the important information (e.g., analysis method, results for the other DVs).   

B. Introduction

  1. The reviews of the negative impacts of distancing policies on children's learning performance mental health do not seem relevant to the objective of the study. The authors shall explicitly explain how those adverse consequences influence parents' mental health.
  2. The author examined a wide range of predictors ranging from school policy to parents' sociodemographics. The authors must justify each of the predictors via reviewing the relevant literature or past findings.    

C. Methodology

  1. The study analyzed the data collected from waves 2 through 5 of the Socio-Economic Impacts of COVID-19 Survey. More details are needed for the data collection. For example, it is important to indicate what data were collected in wave 2.
  2. It was reported that the final analytic sample consisted of 1,913 participants. However, only 4% of them completed four waves. Kindly clarify the total sample size.
  3. Suggest reverse-scoring the mental health score so that the result can be interpreted consistently with the results for the other three DVs. Alternatively, remind the readers about the scoring before interpreting and discussing the results for mental health.

D. Results

  1. what does "Spouse Employment: Single" mean? It was not included in line 123.
  2. Suggest reporting factors that are harmful and followed by factors that are beneficial to ease understanding.   
  3. Lines 190-191, "having a child with a diagnosed learning disability (b = -0.387**) were associated with better mental health" Please verify the result. If it is correctly reported, the authors are urged to discuss this counterintuitive finding.
  4. line 228, "work from home occasionally (b = -0.283*) all the time", it seems like the word "and" is missing. Please check.

E. Discussion   

  1. Lines 287-295, it is more suitable to present the contents in the Introduction.
  2. The current version of the discussion can be further improved. The authors are expected to discuss the key findings for each of the DVs. 
  3. The implications of the findings shall be highlighted. The value of the study can be greater if the authors explicitly discuss (and provide a specific example for) utilizing their findings to promote parents' mental health during a pandemic. 

Round 2

Reviewer 1 Report

Well revised

Reviewer 2 Report

I thank the authors for their efforts in addressing my comments. The revision is satisfactory. 

The authors are reminded to rectify the spelling error in the footnote on p. 4.

"Here, having a spouse or partner was combined with spoausal employment"